# IMPLICIT LATENT CAUSAL REPRESENTATION LEARNING THROUGH SOFT INTERVENTIONS

## ABSTRACT

Learning causal representations from observational and interventional data in the absence of known ground-truth graph structures necessitates implicit latent causal representation learning. Implicitly learning causal mechanisms typically involves two categories of interventional data: hard and soft interventions. In real-world scenarios, soft interventions are often more realistic than hard interventions, as the latter require fully controlled environments. Unlike hard interventions, which directly force changes in a causal variable, soft interventions exert influence indirectly by affecting the causal mechanism. In this paper, we tackle implicit latent causal representation learning in a Variational Autoencoder (VAE) framework through soft interventions. Our approach models soft interventions effects by employing a causal mechanism switch variable designed to toggle between different causal mechanisms. In our experiments, we consistently observe improved learning of identifiable, causal representations, compared to baseline approaches.

## 1 INTRODUCTION

Causal representation learning promotes conventional deep learning models from black boxes to gray boxes such that we can obtain robust, transferable, and explainable representations Schölkopf (2019). In contrast, in statistical representation learning, the representations do not generally have a meaningful structure. Although we can represent statistical models with graphical models just as in causal models, the connections in such a graph are volatile and cannot be generalized to different domains. Furthermore, in causal models we can perform interventions and counterfactuals on the variables which can help the model to imagine situations that it has never seen before.

Causal models can be expressed by both graphical models and structural causal models (SCMs) Pearl (2005). A causal graphical model is a Directed Acyclic Graph (DAG) in which there is an edge between variables such as $X \to Y$ if $X$ is a parent of $Y$. We call a variable *endogenous* if it has a parent in the graph and *exogenous* if it has no parents. Exogenous variable usually correspond to uncertainties and noise in the data Pearl et al. (2016). In graphical causal models we can only infer about which variables are causally related to each other but we cannot infer about *how* they are related to each other. SCMs are a way to describe this relationship.

Identifiability of causal models is one of the long-standing questions in this field. Without identifiablity, we cannot claim the learnt representations are causal. To see this, let's consider three random variables $X, Y, Z$ whose joint distribution is factorized as $p(X, Y, Z) = p(Y|X)p(Z|X)p(X)$. According to the faithfulness assumption, the conditional independence in the joint distribution of variables must be entailed in its corresponding graph as well Yu et al. (2020). Therefore, the joint distribution $p(X, Y, Z)$ can be represented by three graphs $Z \to X \to Y$, $Y \to X \to Z$, and $Y \leftarrow X \to Z$ which imply same conditional independence $Z \perp\!\!\!\perp Y|X$. All of these graphical models belong to the same Markov Equivalence Class (MEC). It can be concluded that the Markov condition in graphs is not sufficient for identifying causal models Schölkopf (2019) and without further assumptions or data, we can only learn the MEC of the causal model.

Existing works have made different assumptions about availability of of ground truth causal variables labels Yang et al. (2021), model parameters Ahuja et al. (2023), availability of paired interventional data Brehmer et al. (2022), availability of intervention targets Lippe et al. (2022) to ensure identifiability of causal models.

In a Variational AutoEncoder (VAE) framework, there are generally two approaches for causal representation learning: Explicit Latent Causal Models (ELCMs) and Implicit Latent Causal Models (ILCMs). IN ELCMs, the latents are the causal variables and the adjacency matrix of the causal graph is parameterized and integrated in the prior of the latents such that the prior of latents is factorized according to the Causal Markov Condition Schölkopf et al. (2021). This approach in causal representation learning is highly susceptible to being stuck in local minimums as it is hard to learn representations without knowing the graph, and it is hard to learn the graph without knowing the representations. To the best of our knowledge, ILCMs were first introduced in Brehmer et al. (2022) and they circumvent this chicken and egg problem in causal model discovery by using *solution functions* which can implicitly model edges in the causal graph. in ILCMs the latents are the exogenous variables and the there is no explicit parameterization for the graph. Let's consider a simple SCM to understand the motivation behind solution functions:

$$Z_1 = f_1(E_1) \quad \& \quad Z_2 = f_2(Z_1, E_2), \tag{1}$$

where $e_1$ and $e_2$ are exogenous variables, $f_1$ and $f_2$ are the causal mechanisms, $z_1$ and $z_2$ are the causal variables. This example SCM is a general case in which the causal variables are an arbitrary nonlinear function of parents and exogenous variables. It is easy to see that we can write all the causal variables as a function of exogenous variables:

$$Z_1 = f_1(E_1) = s_1(E_1, E_2) \quad \& \quad Z_2 = f_2(f_1(E_1), E_2) = s_2(E_1, E_2), \tag{2}$$

where $s_1$ and $s_2$ are the solution functions. This example shows how exogenous variables can be mapped to causal variables through solution functions and that we feed all of the exogenous variables to the corresponding solution function of a causal variable and let the model decide which of the exogenous variables are ancestral. One might wonder why exogenous variables are chosen to be the latents in ILCMs and the reason is that exogenous variables must be mutually independent in a SCM, hence, we can conveniently factorize prior of exogenous variables as $p(E) = \Pi_i p(E_i)$. The observed variables are obtained from from a mixture $g$ of causal variables $Z$ i.e. $X = g(Z)$.

Motivated by the results in Brehmer et al. (2022), we propose a novel approach for implicit causal representation learning with *soft interventions*. In implicit causal representation learning the problem is to recover the exogenous variables $E$ from the observed variables $X$. In Brehmer et al. (2022), the interventions are assumed to be hard which is not realistic and does not apply to real-world problems Correa & Bareinboim (2020). In hard interventions, the connections of the intervened variables with its parents are severed Pearl et al. (2016) whereas in soft interventions, the causal variable is still affected by its parents and the intervention changes the conditional distribution $p(Z_i|Z_{pa})$ Correa & Bareinboim (2020) which includes as a special case the hard interventions $do(Z_i) = z_i$. Our focus here will be on identifiability of causal models and not scalability with number of variables. As also indicated in Brehmer et al. (2022), a method for identifiability can fail with large number of variables even with hard interventions. Nonetheless, we will evaluate implicit causal models in a large real-world dataset, aiming to highlight promising future directions in implciit causal representation learning.

**Contributions** Our key contributions can be summarized as follows:
**I.** A novel approach for implicit causal representation learning with soft interventions.
**II.** Employing causal mechanisms switch variable to model the effect of soft interventions.

## 2 RELATED WORKS

The problem of causal representation learning has gained a lot of attention recently Schölkopf et al. (2021); Kaddour et al. (2022). The major challenge in this problem is identifiability which cannot be simply achieved

by the Markov condition in graphs Schölkopf (2019), hence, further data and assumptions are required. In general, causal representation learning methods can be categorized into Explicit Latent Causal Models and Implicit Latent Causal Models. Our work focuses on identifiability of causal models while retaining implicit modeling and builds on Brehmer et al. (2022).

## 2.1 Explicit Latent Causal Models

CausalVAE Yang et al. (2021) proposes a causal layer in VAE which takes independent exogenous variables as input and maps them into a space in which variables are causally related. The dependent variables are taken by the decoder to reconstruct the inputs. The acyclicity constraint introduced in Zheng et al. (2018) is used here for a continuous optimization. CITRIS Lippe et al. (2022) is a VAE-based framework for identifiying multi-dimensional causal factors from temporal sequences of images where interventions with known targets may have been performed. Therefore, both observational data and interventional data with known targets is required. CITRIS is a variational autoencoder that assigns latent variables to their corresponding causal factors and induces disentanglement by conditioning each latent's prior distribution only on its respective intervention target. It also disentangles autoencoder representations by using normalizing flows. iCaRL leverages data from multiple environments to learn representations. An environment variable $E$ is introduced which can account for changing factors in each environment same as in Bagi et al. (2023). DEAR Shen et al. (2022) addresses the unidentifiability of mutually independent latent factors and use annotated labels of the groundtruth factors and graph structure as supervisory signals. Identifiability of causal models based on available data and various assumptions about the parameters of the decoder and latent space has been studied in Ahuja et al. (2023). Aforementioned methods are score-based Yu et al. (2020) but, identifiability of causal models using maximal ancestral graphs and statistical tests has also been studied in Jaber et al. (2020).

Leveraging Sparse Mechanisms Shift Schölkopf et al. (2021) and sparsity of the causal graphs is proposed in Perry et al. (2022) as an inductive bias which is then used in Lachapelle et al. (2022) for disentangled causal representation learning.

## 2.2 Implicit Latent Causal Models

Existing methods learn the causal graph along with other parameters of the network, which is a difficult optimization problem with no information about causal variables or the graph available . Motivated by their results, our work focuses on identifying causal models using *soft* interventions instead of *hard* interventions as in Brehmer et al. (2022) to bring ILCMs one step closer to real-world applications.

In Brehmer et al. (2022), identifiability is proven theoretically for hard interventions, however, the results for causal models with more than 10 variables indicate that in practice, complex causal models include more ambiguous relations and confounding factors in which case identifying causal model is not straightforward. Identifiability becomes more difficult when we use soft interventions instead of hard interventions. In soft interventions, the effect of causal variables on the observed variables is not clear as both intervention and parents are influencing the causal variables. Also note that, we are retaining implicit modeling, hence, there is no knowledge of parents. In the subsequent sections, we will examine where the identifiabiltiy theory in Brehmer et al. (2022) fails when we use soft interventions and how we can model their effect using causal mechanisms switch technique as introduced in Pearl (2005); Schölkopf (2019) to help model find the true causal model.

## 3 Methodology

### 3.1 Objective

A causal model is a conceptual or mathematical representation used to understand and describe the relationships between different variables and how they influence each other i.e., causal mechanisms. A causal model in its most general form can be expressed as Schölkopf (2019):

$$Z_i = f_i(Z_{PA_i}, E_i) \quad (i = 1, ..., n),\tag{3}$$

where $f_i$ is a deterministic function that defines the relation between causal variable $Z_i$ and its corresponding parents $PA_i$ and exogenous variable $E_i$. In implicit modeling, the latents are the exogenous variables $E_i$ and we aim to obtain causal variables through solution functions $s_i : E \to Z_i$ to circumvent modeling the adjacency matrix of the causal model. Since exogenous variables are assumed to be mutually independent in a causal model, we can conveniently factorize $p(E) = \Pi_i p(E_i)$ whereas prior of causal variables should be factorized according to the Causal Markov condition Schölkopf et al. (2021).

In implicit causal representation learning, the challenge lies in uniquely determining the causal relationships between variables from observed data, a problem known as *Identifiability*. Identifying causal models from data can be complex, often studied within classes of models, such as those identifiable up to affine transformations. In the context of nonlinear Independent Component Analysis (ICA), the generative process involves a mixture $g$ of unobserved latent causal variables $Z \in \mathbb{R}^n$, resulting in observations $X \in \mathbb{R}^n$ i.e. observations Lachapelle et al. (2022); Zheng et al. (2022). Our objective in this paper is to recover $Z$ from $X$."

## 3.2 Setup

Identifying a causal model from observational data is not trivial and requires strict assumptions on the parameters of the model Ahuja et al. (2023). Interventional data can be used along with observational data to observe the effect of changing a causal variable in the observed variables and thereby identifying it. One simple setting here could be to use pair of observed variables $(x, \tilde{x})$ where $x$ is the observational data and $\tilde{x}$ is the interventional data in which one of the causal variables is targeted by a *soft intervention*. In order to simplify the problem we make two assumptions in the generation process:

**Assumption 3.1** (Atomic interventions). The interventional data $\tilde{x}$ is a result of single soft intervention:
$$\tilde{x} = g(z_{/i}, \tilde{z}_i)$$

**Assumption 3.2** (Counterfactual exogenous variables). The soft intervention affects only the exogenous variable of the intervened variable while maintaining the exogenous of all other causal variables:
$$\begin{cases} e_i \neq \tilde{e}_i & if \quad intervention \, on \, z_i \\ e_i = \tilde{e}_i & otherwise. \end{cases}$$

Without these two assumptions the observed changes in $\tilde{x}$ would be ambiguous as multiple interventions or change in exogenous variables might have an overlapping effect in the observed variables. Consequently, the data generation process can be defined as:

**Definition 3.1** (Data generation process). Consider a latent causal model where the underlying SCM is defined by a continuous latent space $E \in \mathbb{R}^n$ with independent probabilities $p(E_i)$ which admits a solution function $s$. The paired observations $(x, \tilde{x})$ are generated as:

$$e \sim p(E), \quad z = s(e), \quad x = g(z), \quad \xrightarrow{\text{Soft-Intervention.}} \tilde{e}_i \sim p(\tilde{E}_i), \quad \tilde{z} = \tilde{s}(\tilde{e}_i, e_{/i}), \quad \tilde{x} = g(\tilde{z}) \quad (4)$$

## 3.3 Proposed method: SoftILCM

The ILCM model from Brehmer et al. (2022) assumes hard interventions, making post-intervention representations independent of ancestry. However, this is limited to controlled scenarios. In response to real-world complexity, we introduce SoftILCM. We address the challenges of identifiability with soft interventions and present a model using a causal mechanism switch variable $n \in \mathbb{R}^n$ to effectively capture soft interventions.

**Definition 3.2** (LCM equivalence up to permutation and reparameterization (informal)) Brehmer et al. (2022)). Let $M = (C, X, g)$ and $M' = (C', X, g')$ be two Latent Causal Models (LCMs) with same observational space $X$, $C$ and $C'$ be SCM, and $g$ and $g'$ be mixing functions. We say $M$ and $M'$ are equivalent up to permutation and reparameterization if there exists a permutation $\psi : G(C) \to G(C')$ ($G$ is graph over $C$) and elementwise diffeomorphisms $\phi : C \to C'$ to map causal and exogenous variables in $C$ to their corresponding ones in $C'$.

**Definition 3.3** (Disentanglement Lachapelle et al. (2022)). Given a groundtruth model $M^*$, we say a learned LCM $M$ is disentangled when $M^*$ and $M$ are equivalent up to permutation and reparameterization.

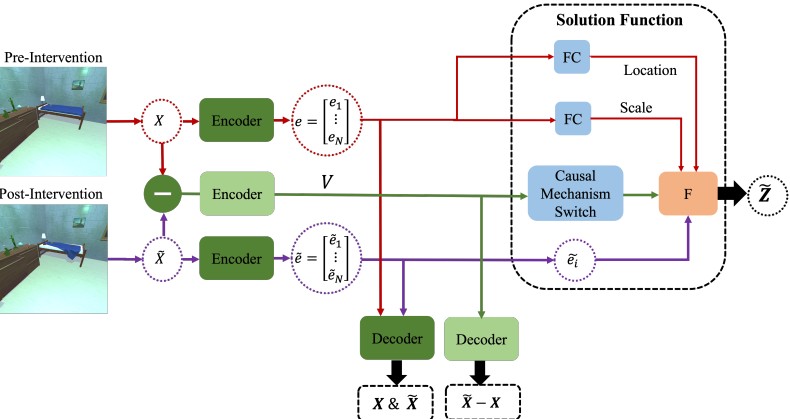

Figure 1: General overview of the SoftILCM. FC represents fully connected neural network, and F shows solution function.

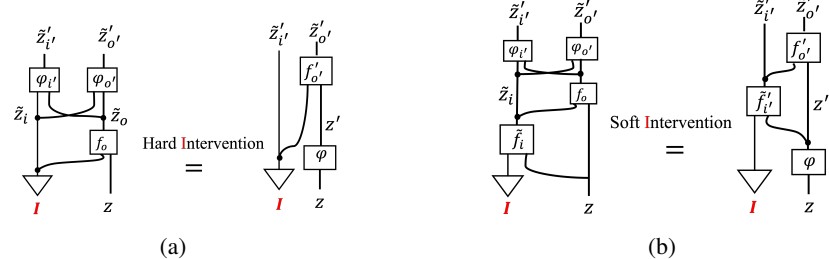

(a)  (b)

Figure 2: Causal graph models in the presence of Hard (a) and Soft (b) interventions.

**Hard intervention:** We begin by examining identifiability in ILCMs according to Definition 3.2 with *hard interventions*. The string in Figure 2a illustrates how post intervention causal variables in $M$ can be mapped to $M'$ if there exists a diffeomorphism $\phi$. It is noteworthy that the intervened variable $\tilde{Z}_i$ has no connection with its parents and is only affected by intervention $I$ as $\tilde{z}_i = f_i(., \tilde{e}_i)$.

**Soft intervention:** In soft interventions, the string diagram will be as depicted in Figure 2b. The major difference here is that $\tilde{Z}_i$ is no longer disconnected from its parents and its causal mechanism $f_i$ is affected by the intervention. One of the conditions for identifiability, is that $\phi$ should be $\psi - diagonal$ Brehmer et al. (2022) which will be violated here and $\tilde{Z}'_{i'}$ will no longer be a function of $\tilde{Z}_i$ alone. Consequently, the causal variables will no longer be disentangled as $\tilde{z}_i = \tilde{f}_i(z_{pa_i}, \tilde{e}_i)$.

**Causal mechanism switch variable:** In implicit modeling we lack knowledge regarding the true parents and the structure of chains and forks within a graph which can induce indirect dependencies. This is attributed to the presence of unconditional dependence inherent in the configurations of chains and forks. Consequently, discerning causal relations from soft interventions becomes profoundly challenging, especially within complex causal models marked by confounding factors and ambiguous relations. To tackle this issue, we propose to model the effect of soft interventions as $\tilde{z}_i = \tilde{f}_i(z_{pa_i}, \tilde{e}_i) = h_i(z_{pa_i}, \tilde{e}_i, v)$, where $v$ is the *causal mechanism switch variable*. Thus, the new parent set of $Z_i$ will be $\tilde{PA}_i = PA_i \cup V$ and it is related to $Z_i$ by the conditional probability:

$$p(z_i|\tilde{pa}_i) = \begin{cases} p(z_i|pa_i, v) & if \quad z_i \; is \; target \; of \; soft \; intervention \\ p(z_i|pa_i) & if \quad no \; interventions \end{cases} \quad (5)$$

The usage of $v$ in soft interventions is analogous to augmented networks in Pearl (2005) which was mainly designed for hard interventions, however, it can also be used for soft interventions:

**Quote:** *"One advantage of the augmented network representation is that it is applicable to any change in the functional relationship $f_i$ and not merely to the replacement of $f_i$ by a constant."* Pearl (2005)

**Theorem 3.4** (Identifiability of $\mathbb{R}$-valued ALCMs from soft interventions-Extended from Brehmer et al. (2022)). *Let $M = (C, X, g, V)$ and $M' = (C', X, g', V')$ be two Augmented Latent Causal Models (ALCMs) with the following properties:*

*1. The ALCMs have same observation space $X$.*
*2. The SCMs $C$ and $C'$ consist of $n$ real valued endogenous and exogenous variables such that $Z_i = E_i = Z'_i = E'_i = \mathbb{R}$.*
*3. The switch variables are real valued $V = \mathbb{R}^n$.*
*4. The endogenous variables are deterministic function of their augmented parents $\tilde{PA} = PA \cup V$.*
*Then the following two statements are equivalent:*

1. *The ALCMs assign same likelihood to interventional and obsevational data i.e., $p_M^X(x, \tilde{x}) = p_{M'}^X(x, \tilde{x})$.*

2. *The ALCMs are equivalent according to Definition 3.2.*

The main part of the proof of theory is to show that the transformation $\phi$ in Figure 2b is $\psi - diagonal$ which is discussed in Appendix A.1.

Consequently, there will be three latents in SoftILCM: a switch variable $V$, the pre-intervention exogenous variables $E$ and the post-intervention causal variables $\tilde{E}$. We would have maximized log-likelihood of $\log p(x, \tilde{x})$ but, as the likelihood is intractable due to the latents, we are going to maximize the Evidence Lower BOund (ELBO) given as follows:

$$\max_{p,q} E_{p^*(x,\tilde{x})} \left[ E_{q(n|\tilde{x}-x),q(e|x),q(\tilde{e}|\tilde{x})} \left[ \log \frac{p(x|e)p(\tilde{x}|\tilde{e})p(\tilde{x}-x|v)p(\tilde{e}|e,v)p(v)p(e)}{q(e|x)q(\tilde{e}|\tilde{x})q(v|\tilde{x}-x)} \right] \right] \quad (6)$$

The approximate posterior $q(.|.)$ of the latents are factorized as we are using mean-field variational inference Goodfellow et al. (2016) and the prior of latents are factorized as $p(\tilde{e}, e, v) = p(\tilde{e}|e, v)p(v)p(e)$. As exogenous variables are mutually independent $p(e) = \Pi_i p(e_i)$ and we assume $p(e_i)$ and $p(v)$ to be standard Gaussian. Furthermore, as we assume $e_i = \tilde{e}_i$ for all non-intervened variables, the $p(\tilde{e}|e, v)$ will be as follows:

$$p(\tilde{e}|e,v) = \Pi_{i \notin I} \delta(\tilde{e}_i - e_i) \Pi_{i \in I} p(\tilde{e}_i|e_i, v) = \Pi_{i \notin I} \delta(\tilde{e}_i - e_i) \Pi_{i \in I} p(\tilde{z}_i|e_i, v) |\frac{\partial \tilde{z}_i}{\partial \tilde{e}_i}| \quad (7)$$

The last equality is obtained from the Change of Variable Rule in probability theory as we use normalizing flows to map $\tilde{e}_i$ to $\tilde{z}_i$ Dinh et al. (2017). The solution function $\tilde{s}_i$ is used to map exogenous variable to causal variables and we implement them using location-scale noise models Immer et al. (2023) as also practiced in Brehmer et al. (2022). For simplicity, in our experiments, we are only going to change the $loc$ network in post-intervention. Thereofore, to satisify the cinditions in Theory 3.4 we are going to use switch variable $V$ as follows:

$$\tilde{z}_i = \tilde{s}_i(\tilde{e}_i; e, v) = \frac{\tilde{e}_i - (loc(e_{/i}) + h(v))}{scale(e_{/i})}, \quad (8)$$

Through this solution function, the model implicitly learns which of $e_{/i}$ are ancestral exogenous variables, hence, identifying causal relations. We assume $p(\tilde{z}_i|e, v)$ to be a Gaussian whose mean is determined by $e_i$ and $n$ as $e_i$ is a certain ancestor of $\tilde{z}_i$ and we do not have knowledge about the other ancestors in implicit

modeling. We also obtain $v$ from $\tilde{x} - x$ to guide the model that $v$ should model effects of soft intervention as we assume the changes in $\tilde{x}$ are due to the soft intervention only. The general overview of the model is illustrated in Figure 1.

## 4    EXPERIMENTS AND RESULTS

The experiments conducted in this paper address two downstream tasks; (1) Causal Disentanglement to identify the true causal graph from pairs of observations $(x, \tilde{x})$, and (2) Action Inference to infer about actions generated from the post-intervention samples using causal variables in a supervised manner.

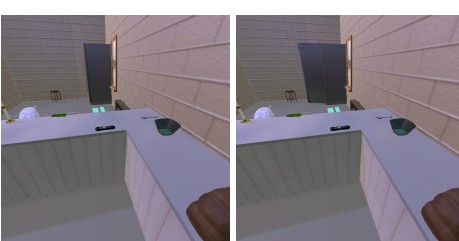 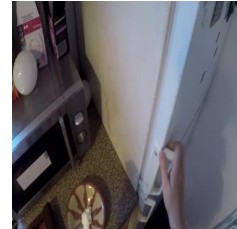 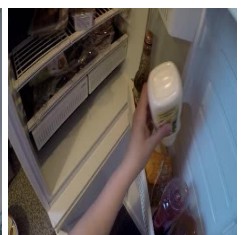

|  (a) Pre-Intervention | (b) Post-Intervention | (c) Pre-Intervention | (d) Post-Intervention |

Figure 3: Image samples of Pre and Post interventions in simulated (a, b), as well as real-world (c, d) conditions from Causal-Triplet datasets Liu et al. (2023).

### 4.1    DATASETS

#### 4.1.1    SYNTHETIC

We generate simple synthetic datasets with $X = Z = \mathbb{R}^n$. For each dimension $n$, we generate ten datasets, using linear SCMs with random DAGs and causal mechanisms, in which each edge is sampled in a fixed topological order from a Bernoulli distribution with probability 0.5. The pre-intervention ans post-intervention causal variables are obtained as:

$$z_i = scale(z_{pa_i})e_i + loc(z_{pa_i}) \xrightarrow{\text{Soft-Intervention.}} \tilde{z}_i = s\tilde{cale}(z_{pa_i})\tilde{e}_i + \tilde{loc}(z_{pa_i}), \qquad (9)$$

where the $loc$ and $scale$ networks are changed in post intervention. The pre-intervention $loc$ and post-intervention $\tilde{loc}$ network weights are initialized with samples drawn from $N(0, 1)$ and $N(3, 1)$, respectively. The $scale$ is 1 (no effect from parents) for both pre-intervention and post-intervention samples Both $e_i$ and $\tilde{e}_i$ are sampled from a standard Gaussian. The causal variables are mapped to the data space through a randomly sampled $SO(n)$ rotation. For each dataset, we generate 100,000 training samples, 10,000 validation samples, and 10,000 test samples.

#### 4.1.2    CAUSAL-TRIPLET

Causal-Triplet datasets Liu et al. (2023) consist of: images obtained from a photo-realistic simulator of embodied agents, ProcTHOR Deitke et al. (2022), and the other contains images repurposed from a real-world video dataset of human-object interactions Damen et al. (2022). The former one contains 100k images in which 7 types of actions manipulate 24 types of objects in 10k distinct ProcTHOR indoor environments. The latter consists of 2632 image pairs, collected under a similar setup from the Epic-Kitchens dataset with 97 actions manipulating 277 objects. Samples of both datasets are shown in Figure 3

### 4.2    METRICS

For the causal disentanglement task, we are going to use the DCI scores Eastwood & Williams (2018). Causal disentanglement score quantifies the degree to which $Z_i$ factorises or disentangles the $Z^*$. Causal disentanglement $D_i$ for $Z_i$ is calculated as $D_i = (1 - H_K(P_{i.})) = (1 + \sum_{k=0}^{K-1} P_{ik} \log_K P_{ik})$ where $P_{ij} = \frac{R_{ij}}{\sum_{k=0}^{K-1} R_{ik}}$ and $R_{ij}$ denotes the probability of $z_i$ being important for predicting $z_j^*$. Total causal disentanglement is the weighted average $\sum_i \rho_i D_i$ where $\rho_i = \frac{\sum_j R_{ij}}{\sum_{ij} R_{ij}}$. Causal Completeness quantifies

the degree to which each $z_i^*$ is captured by a single $Z_i$. Causal completeness is calculated as $C_j = (1 - H_D(\tilde{P}_{\cdot j})) = (1 + \sum_{d=0}^{D-1} \tilde{P}_{dj} \log_D \tilde{P}_{ij})$. $D$ and $K$ here are equal to the dimension of $z^*$ and $z$ which is $n$.

For the action inference task, we will use classification accuracy as a metric.

## 5 RESULTS

### 5.1 CAUSAL DISENTANGLEMENT

We generated a dataset for the soft interventions and trained the models of SofTILCM, ILCM, $\beta$-VAE and D-VAE for 10 different seeds, which generated 10 different causal graphs. We selected 4 causal variables to encompass complex causal structures, including forks, chains, and colliders. Table 1 displays the Causal Disentanglement and Causal Completeness Scores for the three baseline models, computed on the test data.

Table 1: Comparison of identifiability results

| Graph | | Causal Disentanglement | | | | Causal Completeness | | | |
|---|---|---|---|---|---|---|---|---|---|
| Model | Name | $\beta$-VAE | $d$-VAE | ILCM | Soft-ILCM | $\beta$-VAE | $d$-VAE | ILCM | Soft-ILCM |
|  | G-1 | 0.39 | 0.54 | 0.69 | **0.81** | 0.53 | 0.68 | 0.76 | **0.87** |
|  | G-2 | 0.15 | 0.72 | 0.74 | **0.83** | 0.24 | 0.77 | 0.79 | **0.87** |
|  | G-5 | 0.25 | 0.51 | 0.67 | **0.97** | 0.54 | 0.56 | 0.77 | **0.97** |
|  | G-6 | 0.19 | 0.50 | 0.65 | **0.69** | 0.41 | 0.69 | 0.76 | **0.79** |
|  | G-7 | 0.25 | **0.44** | 0.32 | 0.42 | 0.45 | **0.54** | 0.37 | 0.50 |
|  | G-8 | 0.51 | 0.62 | 0.73 | **0.98** | 0.63 | 0.69 | 0.73 | **0.98** |
|  | G-9 | 0.35 | 0.49 | 0.70 | **0.76** | 0.65 | 0.74 | 0.88 | **0.90** |
|  | G-10 | 0.48 | 0.54 | 0.50 | **0.60** | 0.61 | 0.63 | 0.62 | **0.69** |
|  | G-11 | 0.31 | 0.68 | 0.83 | **0.86** | 0.40 | 0.76 | 0.86 | **0.87** |
|  | G-12 | 0.15 | 0.39 | **0.52** | 0.32 | 0.42 | 0.56 | **0.82** | 0.50 |

The results in Table 1 indicate that SoftILCM can identify the true causal graph in most cases. The results for graphs $G - 7$ and $G - 12$ are worst and there can be a seen a pattern. As mentioned in Schölkopf et al. (2021); Perry et al. (2022), causal graphs are sparse and in $G - 7$ case where the graph is fully connected the proposed method cannot identify the causal variables well. Furthermore, in the next experiment we are going to show that the causal disentanglement has a lower bound which is a function of number of edges in the graph and the intensity of soft intervention effect which can explain why SoftILCM cannot identify causal variables in $G - 12$ despite its sparsity.

### 5.2 FACTORS AFFECTING CAUSAL DISENTANGLEMENT

In this experiment, we consider the graph $G - 5$ which has the best identifiability and change the intensity of soft intervention and number of edges in its data generation process. For changing intensity, the post-intervention $\tilde{loc}$ network weights are initialized with samples drawn from $N(1, 1)$ (almost similar to $loc$)

and $N(10,1)$ (significantly different from $loc$). For changing number of edges, we consider a chain and fully-connected graph.

Table 2: SoftILCM performance on different configurations of $G-5$

| Edges | Post-intervention causal mechanism | Causal Disentanglement | Causal Completeness |
|---|---|---|---|
| Chain | Default | 0.98 | 0.98 |
| Full | Default | 0.89 | 0.89 |
| Default | Significantly different | 0.68 | 0.73 |
| Default | Almost similar | 0.85 | 0.86 |

The results in Table 2 further confirms the sparsity of causal graphs as the causal disentanglement is much worse in the fully-connected graph than the default graph of $G-5$. The result for significantly different post-intervention causal mechanism indicate that the switch variable cannot approximate intense effects of soft intervention as discussed in Theory 3.4 and is not a determinsitc function of $V$ and its parent. Almost similar post-intervention causal mechanisms also do not have sufficient variability to disentangle the causal variables as studied in Lachapelle et al. (2022).

### 5.3 ACTION INFERENCE

In this experiment, we show the performance of SoftILCM in a real-world Causal-Triplet datasets. Based on the nature of actions in this dataset, the causal variables should represent attributes of objects such as shape and color. As the dataset is consisted of images we train all the methods with ResNet encoder and decoder. One of the constraints of ILCM and SoftILCM is to have a diffeomorphic encoder. To ensure this, we add the following inverse consistency constraint to the loss function to ensure that decoder is the inverse of encoder as: $\sum_i E_{e_i \sim q(e_i|x), \hat{e}_i \sim q(\hat{e}_i|\hat{x})}[(e_i - \hat{e}_i)^2]$, where $\hat{x}$ are the reconstructed samples. For the ProcThor dataset the number of causal variables are 7. For the Epic-Kitchens dataset, we randomly chose 20 actions from the dataset as 97 causal variables will be too complex in a VAE setup. The results are shown in Table 3. The results in Table 3 indicate that SoftILCM performs at par with the baseline methods. Based on the number of actions in this dataset, the results suggest that implicit causal models are not scalable to causal models with large number of variables. Furthermore, due to the unconditional dependencies of forks and chains in causal graphs, including all causal variables in the action or object inference may cause spurious correlations. We leave the scalability of SoftILCM to our future works.

Table 3: Action and object accuracy of different methods on Causal-Triplet datasets

| Method | Epic-Kitchens | | ProcTHOR | |
| | Action Accuracy | Object accuracy | Action Accuracy | Object accuracy |
|---|---|---|---|---|
| **ILCM** | 0.17 | 0.18 | 0.21 | 0.34 |
| $\beta-VAE$ | **0.30** | 0.20 | **0.32** | **0.41** |
| $d-VAE$ | 0.19 | 0.20 | 0.27 | 0.38 |
| **Soft-ILCM (ours)** | 0.17 | **0.21** | 0.22 | 0.36 |

## 6 CONCLUSION

Our research introduces SoftILCM, a novel model for implicit latent causal representation learning under soft interventions, incorporating a causal mechanism switch variable. We've conducted evaluations on synthetic and real-world datasets, employing the causal disentanglement score as a key metric. The results demonstrate that SoftILCM outperforms state-of-the-art methods, highlighting its efficacy in causal representation learning and moving implicit causal models one step towards the real-world applications. Our results indicate that SoftILCM can identify the causal models from soft interventions with few number of variables and it can be a promising direction for future works.

## 7 ETHICS STATEMENT

In accordance with the principles set forth in the ICLR Code of Ethics, we wish to affirm that this research paper does not raise any ethical concerns. The design, execution, and reporting of our research adhere to the ethical standards outlined by the ICLR. We have taken all necessary precautions to ensure the responsible conduct of research, including considerations of fairness, transparency, and the protection of individuals' rights and privacy.

## 8 REPRODUCIBILITY STATEMENT

In adherence to transparency and reproducibility standards, we have provided all the necessary code in the supplementary material to replicate the results presented in the main draft of this paper. Detailed instructions and documentation required to reproduce the experiments are also included alongside the code.

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

# A   APPENDIX

## A.1   PROOF OF IDENTIFIABILITY THEORY

**Proposition A.1** (Measure preservation of causal mechanisms with switch variable). Let $Z \in \mathbb{R}^n$ be pre-intervention causal variables and $V \in \mathbb{R}^n$ be the causal mechanisms switch variable. The post intervention causal variables can then be formulated as: $\tilde{z}_i = \tilde{f}_i(z_{pa_i}, \tilde{e}_i) = h_i(z_{pa_i}, \tilde{e}_i, v)$. In implicit modeling, as we do not know the parents, we feed all $Z$ to causal mechanisms, hence, $h_i : V \times Z \to \tilde{Z}_i$. To explore the properties of $h_i$ under this transformation, define conditional probability measure on $p_{\tilde{Z}_i|Z}$ on $\tilde{Z}_i$ and probability measure $p_V$ on $V$. For all $z \in Z$, the function $h_i(., z) : V \to \tilde{Z}_i$ preserves the measure on $V$ and induces a consistent conditional distribution $p(\tilde{Z}_i|Z)$.

The proof of Proposition A.1 is straightforward when assuming that the value of $\tilde{Z}_i$ is a deterministic function of $V$ and $Z_{pa_i}$ with no uncertainties. However, in reality, the causal mechanism is represented as a function of $V$ and $V_{pa_i}$, approximating the impact of soft intervention, and therefore, zero uncertainties are not guaranteed. Following this proposition and Lemma 2 from Brehmer et al. (2022), we can prove the identifiability of implicit causal models:

**Corollary A.1** ($\psi - diagonal$ diffeomorphic Transformation). The post-intervention causal mechanisms $\tilde{f}$ are constant in pre-intervention causal variables $Z \in \mathbb{R}^n$, hence, there exists a diffeomorphic transformation $\phi = g'^{-1} \circ g : Z \to Z'$ which is $\psi - diagonal$.

Using Corollary A.1, the rest of the proof for identifiability follows from Brehmer et al. (2022). As we will demonstrate in the results section, the causal mechanisms switch approach for approximating post-intervention causal mechanisms may not yield ideal results in certain instances.

## A.2   SOFT VS. HARD INTERVENTION

In a causal model, an intervention refers to a deliberate action taken to manipulate or change one or more variables in order to observe its impact on other variables within the causal model. Interventions help to study how changes in one variable directly cause changes in another, thereby revealing causal relationships.

Based on the levels of control and manipulation in a causal intervention, we can have soft vs. hard interventions. A hard intervention involves directly manipulating the variables of interest in a controlled manner such as Randomized Controlled Trials (RCTs). In other words, a hard intervention sets the value of a causal variable $Z$ to a certain value denoted as $do(Z = z)$ Pearl et al. (2016).

On the other hand, soft intervention involves more subtle or less controlled manipulation of variables and changes the conditional distribution of the causal variable $p(Z|Z_{pa}) \to \tilde{p}(Z|Z_{pa})$ which can be modeled as $\tilde{z}_i = \tilde{f}_i(z_{pa_i}, \tilde{e}_i)$ Correa & Bareinboim (2020).

Looking at interventions from a graphical standpoint, a hard intervention entails that the intervened node is solely impacted by the intervention itself, with no influence coming from its ancestral nodes. Conversely, in the context of a soft intervention, the representation of the intervened node can be influenced not only by the intervention but also by its parent nodes.

As an example, suppose we are trying to understand the causal relationship between different types of diets and weight loss. The *soft intervention* in this scenario could be a switch from a regular diet to a low-carb diet. Switching to a low-carb diet is a voluntary choice made by the individual and there are no external forces or regulations compelling them to make this change (non-coercive). The intervention involves a modification of the individual's diet rather than a complete disruption since they are adjusting the proportion of macronutrients (fats, proteins, and carbs) they consume, which is less disruptive than a radical change in eating habits (gradual modification). The individual has autonomy to choose and tailor their diet according to their preferences and health goals so they are empowered to make informed decisions about their dietary choices (behavioural empowerment). Conversely, if the government or an authority were to intervene and enforce a mandatory

low-carb diet through legal means, this would constitute a *hard intervention*. In this scenario, regulations would be implemented, prohibiting the consumption of specific carbohydrate-containing foods. Regulatory agencies would be established to oversee and ensure adherence to the low-carb diet mandate, taking actions such as removing prohibited foods from the market, restricting their import and production, and so on. Individuals caught consuming banned foods would be subject to fines, legal repercussions, or other penalties.

## A.3 EXPERIMENTS

This section contains additional details about Soft-ILCM design architectures as well as experiments settings.

### A.3.1 ARCHITECTURE DESIGN

Based on the Soft-ILCM architecture depicted in Figure 1, we devised a solution function comprising three fully connected networks. These networks consist of two layers each, with 64 hidden units per layer and ReLU activation functions. For our synthetic dataset experiments, we employed a fully connected auto-encoder model, while for the realistic dataset, we utilized ResNet-based networks.

## A.4 RESULTS

In our synthetic dataset experiments, Soft-ILCM achieved the highest average performance with a 4-variable causal model. We employed various data generation seeds, resulting in diverse adjacency matrices, reflecting the influence of soft interventions on causal mechanisms and parental impact.

While our primary research objective centered on addressing identifiability challenges in implicit causal models under soft interventions, we also conducted an investigation into the scalability of our proposed model. To comprehensively assess its performance, we designed experiments covering a range of causal graphs, featuring 5 to 10 variables, with 10 different seeds for each variable, following a similar experimental setup as our 4-variable causal graph experiments. The outcomes of these experiments, comparing Soft-ILCM and ILCM, are presented in Figure 4.

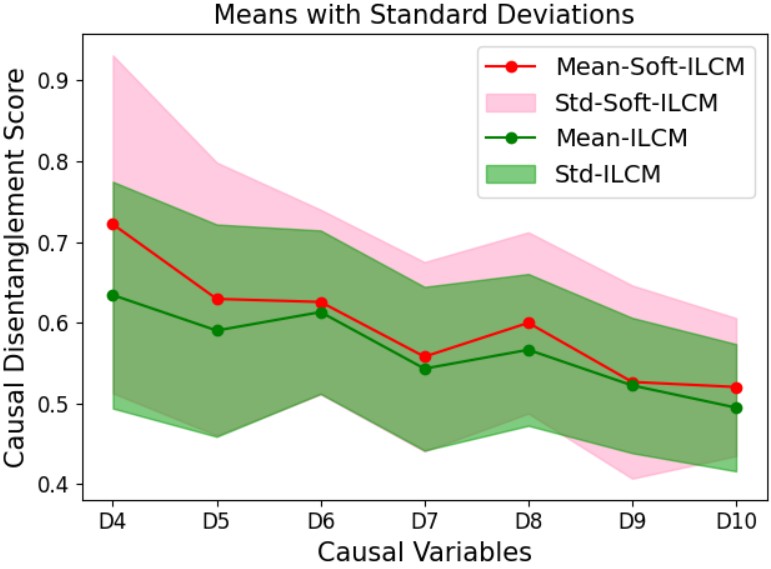

Figure 4: Causal disentanglement for different number of variables

