# OpenReview forum: "Implicit Latent Causal Representation Learning through Soft Interventions"
_ICLR.cc/2024/Conference — ICLR 2024 Conference Withdrawn Submission_

### Official Review · Reviewer_mX9b · 2023-10-31

**Soundness:** 1 poor
**Presentation:** 2 fair
**Contribution:** 2 fair
**Rating:** 3
**Confidence:** 5

**Summary:**

In recent years, the development of theory and methods for identification of causal representations has come into significant prominence. One family of methods focuses on weak supervision derived from observation of data pre and post interventions to learn causal representations.  In the work of Brehmer et al., which extends the work of Locatello et al., the authors developed comprehensive theory and methods for identification in the presence of data of the form pre and post intervention.  In Brehmer et al., the authors worked with hard interventions but the question of how to tackle soft interventions remained open. In this work, the authors extend the framework of Bremer et al. They develop theory and methods for soft interventions. A key component of their proposal is the introduction of switch mechanism to model soft interventions. The authors also validate their proposal on some semi-synthetic datasets.

**Strengths:**

The authors tackle an important and non-trivial extension of Brehmer et al. A successful extension would have useful theoretical and empirical implications. The authors present an interesting approach that relies on inferring a latent switch variable that models the impact of soft intervention.

**Weaknesses:**

I have several major concerns with the paper.

1. In equation (5), authors introduce switch variable v to model soft interventions. This way of modeling soft interventions excludes a large class of soft interventions. Under a soft intervention, the mechanism that generates the intervened variable is changed. Under the new mechanism, the set of parents can change. In equation (5), the only permitted soft interventions are those in which the new variables are added to the set of parents. How do you take care of the case where under the intervention some of the variables in the parent set are dropped?

2. In the two paragraphs that are below Figure 2, the authors already acknowledge that the tilde{z} is entangled and approach from Brehmer won’t extend to soft interventions. However, later with introduction of switch mechanism they don’t make it clear how it takes care of the challenges posed in the setting of Brehmer. The proof in Brehmer crucially used perfect intervention. In the proof overview provided in the main body of Brehmer et al.,  they use perfect interventions in Step 2 in the main body of their work. It is not clear how does the introduction of switch variable circumvent the troubles.

3.  The proof of the main results (Proposition A.1) is not clear and seems incorrect.  The authors state in the Appendix that
“The proof of Proposition A.1 is straightforward when assuming that the value of \tilde{Z}_i is a deterministic function of V and Zpai with no uncertainties. However, in reality, the causal mechanism is represented as a function of V and Vpai , approximating the impact of soft intervention, and therefore, zero uncertainties are not guaranteed. Following this proposition and Lemma 2 from Brehmer et al. (2022), we can prove the identifiability of implicit causal models:”
I cannot see how this is the proof to the claim. There are two sources of mystery i) are the authors trying to invoke further assumptions by stating that "\tilde{Z}_i is a deterministic function of V and Zpai with no uncertainties. " If so why not state it explicitly in the proposition, ii) even under this assumption it is not clear why doe the proof from Brehmer et al. follows. In the current way the authors state it, the result looks incorrect.

4. The experimental results on Causal Triplet dataset show that beta-VAE scales better than the proposed approach, which is not a very promising sign for the proposal. Do the authors have some thoughts on how to improve this?

Overall, the paper needs significant work, the theorems need to be corrected, the assumptions properly stated. Due to these concerns, on the axis of quality, significance, and clarity the paper performs poorly.

**Questions:**

Please see the weakness section, where I highlight my issues and questions.

---

### Official Review · Reviewer_4MaE · 2023-11-01

**Soundness:** 2 fair
**Presentation:** 2 fair
**Contribution:** 2 fair
**Rating:** 5
**Confidence:** 4

**Summary:**

This paper studies learning latent causal representation implicitly, which means the graphical structure is also parameterized and learned. It shows identifiability results with soft interventions and verifies the claims empirically.

**Strengths:**

The experiments appear to be extensive.

**Weaknesses:**

The format of the paper does not seem right. The space at the bottom is too big?

The related work seems to be lacking. For instance, i-VAE seems to be a reasonable baseline. There are also a lot of work on learning under soft interventions.

The paper is a little hard to follow.

**Questions:**

The paper says in section 3.2 “without these two assumptions the observed changes in $\tilde{x}$ would be ambiguous as multiple interventions or change in exogenous variables might have an overlapping effect in the observed variables”. Can you give an example?

In the definition 3.2, you used diffeomorphims between C and C’ which are SCMs. What does diffeomorphism between SCMs mean?

Can you explain a little more why causal mechanism switch variable is needed to handle chains and forks?

Why is decoder not using Z in figure 1?

---

### Official Review · Reviewer_dD4K · 2023-11-01

**Soundness:** 2 fair
**Presentation:** 2 fair
**Contribution:** 2 fair
**Rating:** 3
**Confidence:** 3

**Summary:**

This manuscript extends the ILCM proposed by Brehmer et al. 2022 to cases with soft intervention. While the considered extension is practical and large-scale real-world experiments have been conducted, there are some aspects that could greatly improve the clarity of the manuscript.

**Strengths:**

1. Compared to hard interventions, it is essential to consider the identifiability of soft interventions.

2. The efficacy of the proposed method has been validated on large-scale real-world image datasets.

**Weaknesses:**

1. Some descriptions make the problem setting unclear. For example, what is the essential difference between the considered setting and ICA? In the last paragraph of the introduction, it is mentioned that “In implicit causal representation learning the problem is to recover the exogenous variables E from the observed variables X”; Similarly, the last sentence of Sec. 3.1 states that “Our objective in this paper is to recover Z from X”. Since exogenous variables are independent of each other, the considered problem setting seems to be very similar to ICA. If not, the difference should be clearly stated to distinguish the proposed method from works in nonlinear ICA; if so, there is a lack of discussion on much of the related work on the identifiability of nonlinear ICA—nonlinear ICA with auxiliary variables has been extensively studied, and soft interventions could be seen as a type of auxiliary variable.

2. The manuscript itself is not very self-contained. It requires readers to be very familiar with Brehmer et al. 2022 to get a better sense of the considered problem. For instance, the most crucial concept in ILCMs is referred to as the solution function, which is the mapping from exogenous noise to the causal variable. However, this idea is classical in causal discovery and has been widely used in various algorithms, such as how LiNGAM builds the connection between ICA and causal discovery. As a result, readers might not be fully convinced by the necessity of introducing ILCMs. Thus, more discussion on the motivation of the considered problem is necessary.

3. Since soft interventions can be treated as a type of distributional change, the proposed method should also be discussed in the context of causal discovery with multiple distributions. For example, Huang et al. 2018 introduced CD-NOD, which essentially treats the domain index as an augmented variable for structure learning with distributional changes. It seems that the proposed causal mechanism switch variable is similar to that augmented variable, and more discussion would be helpful.

4. Some definitions are not rigorous. For example, in Definition 3.2, what does an elementwise diffeomorphism on SCM mean? Is C a set of variables and edges? If not, what does “G is a graph over C” mean? Since Definition 3.2 is the exact goal of identifiability, leaving it as “informal” leads to significant confusion.

**Questions:**

Please refer to the above-mentioned confusion regarding some concepts in the manuscript.

---

### Official Review · Reviewer_W75o · 2023-11-01

**Soundness:** 1 poor
**Presentation:** 1 poor
**Contribution:** 1 poor
**Rating:** 3
**Confidence:** 3

**Summary:**

This work presents an extension of the results in (Brehmer et al., 2022) by considering scenarios where weak supervision arises from paired observations of the same system before and after a soft intervention.

**Strengths:**

The paper aims at relaxing one of the assumptions in (Brehmer et al., 2022).

**Weaknesses:**

1. The paper lacks self-containedness, as it heavily depends on the results outlined in (Brehmer et al., 2022), to the extent that it is difficult to pinpoint where exactly the added technical contribution of the present paper lies. For example, for most of the proofs, the reader is referred to (Brehmer et al., 2022). A large part of sec. 3.3 follows very closely sec. 4.2 of (Brehmer et al., 2022). This makes it more challenging to assess the significance of the theoretical claims in the paper, and to confirm their accuracy. Moreover, assumptions and requirements of the proposed framework may not be clearly stated or discussed in the paper (e.g., the required number of atomic interventions; or "variability" as referred to at the end of sec. 5.2).

2. The paper is introduced as a method for causal representation learning based on soft interventions. This is misleading, since it appears that the paper, building on (Brehmer et al., 2022), requires instead weakly supervised (paired) _counterfactual_ data [1]---where the action linking factual and counterfactual scenarios is a soft instead of hard intervention. By presenting it as a method for causal representation learning based on soft interventions, it might instead appear as if the method were closer in spirit to works on interventional causal representation learning [2-6], including those with specific results on soft interventions [7, 8].

3. The Introduction contains a brief review of causality and causal inference, where several parts are ambiguous or unclear. The third paragraph presents the problem of identifiability in CRL in a way which is misleading. The authors discuss Markov equivalence classes: while these play a key role in identifiability of the causal graph based on observational data (i.e., in observational causal discovery), the problem as typically referred to in causal representation learning (also) stems from additional issues of representation learning [9, 10], which also persist in the context of causal representation learning (see, e.g., Fig. 2 in [7]). In the introduction, no references are given for what the authors refer to as "Explicit Latent Causal Models", or for the sentence _"this approach [...] is highly susceptible to being stuck in local minimums"_ (sic, typo: minima). The references presented in sec. 2.1 are not comprehensive. Of particular relevance for the setting discussed in this manuscript are works that require counterfactual data (e.g., [1]), and works which discuss soft interventions (e.g., [7, 8]).

Other issues:

4. The SCM notation (e.g., eq. (1) and following) should be $Z:=f_1(E_1)$. After the equation, there is inconsistent use of uppercase and lowercase letters. Presumably, uppercase represents random variables, whereas lowercase denotes values, but this is not clarified before eq. (1).

5. Sec. 5: _"As mentioned in [...], causal graphs are sparse"_: The sparsity of causal graphs is not an absolute fact, but rather an assumption, though it can be considered a reasonable one in certain situations.

6. The reference to confounding on page 5 seems misleading, as it appears at odds with the assumption of unconditional independence of the exogenous variables stated in the Introduction. It is not clear at all that the present work tackles the issue of _"discerning causal relations"_ within causal models _"marked by confounding"_.

7. The authors write that _"in graphical causal models we can only infer about which variables are causally related to each other but we cannot infer about_ how _they are related to each other"_ (sic, emphasis by the authors). In my view, this sentence does not accurately describe SCMs: causal Bayesian networks also describe how variables are related to each other (depending on the meaning of "how", which is ambiguous), only through conditional probabilities encoding stable causal mechanisms, as opposed to deterministic functions of some parent variables together with exogenous noises, as in SCMs.

References:

[1] Von Kügelgen, Julius, et al. "Self-supervised learning with data augmentations provably isolates content from style." Advances in neural information processing systems 34 (2021): 16451-16467.

[2] Squires, Chandler, et al. "Linear Causal Disentanglement via Interventions." (2023).

[3] Varici, Burak, et al. "Score-based causal representation learning with interventions." arXiv preprint arXiv:2301.08230 (2023).

[4] Ahuja, Kartik, et al. "Interventional causal representation learning." International conference on machine learning. PMLR, 2023.

[5] Buchholz, Simon, et al. "Learning Linear Causal Representations from Interventions under General Nonlinear Mixing." NeurIPS 2023.

[6] von Kügelgen, Julius, et al. "Nonparametric Identifiability of Causal Representations from Unknown Interventions." NeurIPS 2023.

[7] Liang, Wendong, et al. "Causal Component Analysis." NeurIPS 2023

[8] Zhang, Jiaqi, et al. "Identifiability guarantees for causal disentanglement from soft interventions." arXiv preprint arXiv:2307.06250 (2023).

[9] Hyvärinen, A., & Pajunen, P. (1999). Nonlinear independent component analysis: Existence and uniqueness results. Neural networks, 12(3), 429-439.

[10] Locatello, Francesco, et al. "Challenging common assumptions in the unsupervised learning of disentangled representations." international conference on machine learning. PMLR, 2019.

**Questions:**

- Please clarify what is the technical novelty in the theoretical results and proofs of this paper. It seems like all the new proofs for the identifiability results in this work are contained in App. A.1, which contains a Corollary and a Proposition whose proof is "straightforward" under certain assumptions.

- Please provide more explanation on why the Causal-Triplet dataset would be accurately described by your model. One of the stated goals of the paper is to bridge the gap between implicit causal models and their practical applications in real-world scenarios. Nonetheless, there is limited discussion on why the semi-realistic dataset would align with the modeling assumptions of the proposed framework.

- Please clarify what you mean by "causal mechanisms". It is unclear whether the authors refer to it as the functional assignment of an endogenous variable in terms of its parents (endogenous) and its exogenous noise; or whether they refer to the deterministic function expressing a variable in terms of the exogenous noises of all ancestors. Using "causal mechanisms" in this latter sense would be inconsistent with other literature in causal inference.

- It appears as if the solution functions framework cannot distinguish between fully connected and sparse SCMs. My understanding is that in general, there is no way to distinguish whether a variable depends only on its direct causal parents, or on all its ancestors in the graph based on the framework. I would like to ask the authors whether they could comment on this

---

### Official Review · Reviewer_Uh9i · 2023-11-01

**Soundness:** 2 fair
**Presentation:** 2 fair
**Contribution:** 2 fair
**Rating:** 3
**Confidence:** 3

**Summary:**

The paper proposes a method to learn an implicit latent causal model using pairs of $(x, \tilde{x})$ where $x$ is the observational output, and $\tilde{x}$ is the output after a (atomic) soft intervention on the underlying latents $z$. In ILCMs, a solution function that maps independent exogenous variables to the latents $z$ is learned.

**Strengths:**

The paper generalizes learning ILCMs from soft interventions (which allows the intervened variable to continue to depend on its parents). The paper learns the latents by maximizing an ELBO for the evidence $p(x, \tilde{x})$.

**Weaknesses:**

The paper is poorly written and it is difficult to understand the precise assumptions made by for the theoretical results and what identifiability guarantees exist in the paper's setup (more details below). The clarity of the paper would have to be substantially improved and more intuition/explanation needs to be provided for the presented results.

Also, it seems like the paper is formatted incorrectly. The bottom margin seems to be bigger than the default margins. Was this accidental?


### Assumptions on the soft interventions:

The authors model soft interventions using switch variables (Eq. 5). However, in Assumption 3.2, the paper says that the interventions only change the exogenous variable $e_{i}$ on the intervened node $z_i$. This seems to be a restricted kind of soft intervention since only the exogenous node can be changed. For example, this wouldn't allow interventions where the strength or the functional relationship from the endogenous nodes $Z_{\text{pa}(i)}$ changes. Can the authors comment on this? This must be clearly explained and discussed in the paper.
However, in the experiments, more general types of soft interventions were being considered. So I wasn't sure what role this assumption is playing.

### Assumptions for identifiability

The paper does not clearly state any identifiability results for learning the solution functions. What assumptions are needed to identify the solution functions consistently? Do you assume data with soft interventions on each node? Do you assume that the intervention targets are known? It is not clear what assumptions are needed on the pairs $x, \tilde{x}$ to correctly identify the ILCM.

Re Theorem 3.4:
This theorem is provided without much intuition and I found it difficult to understand what it is saying. It seems to be saying that if two ALCMs assign the same density (observational + interventional), then they are equivalent. However, this does not address whether you can identify an equivalent ALCM under your assumptions on the soft interventions.

Moreover, I don't see a proof of this theorem even in the appendix so I couldn't verify the proof either.

### Understanding the ELBO

The paper suggests maximizing the ELBO in Eq. 6. Where does this ELBO come from? I don't see a proof or the derivation in the Appendix. The form of the ELBO is not obvious to me. Where does $p(x - \tilde{x})$ come from? More details need to be provided here. Same comment for Eq. 7.

In Eq. 8, what are `loc` and `scale` and $h(.)$?

Moreover, it seems like further assumptions are being made on the underlying latent distributions when you use a location scale model and a gaussian for $p(v)$?

### Re model architecture:

The architecture is summarized in Fig. 1. But very few details are given about the actual architecture in the paper. So I found even the architecture difficult to understand.

**Questions:**

See Weaknesses section for my questions.

---

### Official Review · Reviewer_sELc · 2023-11-03

**Soundness:** 3 good
**Presentation:** 3 good
**Contribution:** 3 good
**Rating:** 5
**Confidence:** 4

**Summary:**

This paper presents VAE-based implicit causal representation learning under the assumption of soft interventions. This is built on a previous ILCM model (Brehmer et al), but relaxes the assumption of hard interventions to more realistic soft interventions. The core of the method is in the introduction of a switch variable to model the change of causal mechanism due to soft intervention. The experiments were conducted on both synthetic and real datasets, in comparison to ILCM, beta-VAE, and D-VAE as baselines. While the results on synthetic data demonstrated significant improvements in causal disentanglement, the results on real-world datasets seemed to suggest that additional work is needed in order to fully understand the effectiveness and contribution of  the presented method.

**Strengths:**

The paper is overall well written, and the related works well reviewed. The challenges of ILCM are well conveyed, and the main gap to be filled (the assumption of hard interventions) is clearly outlined.

The proposed switching mechanism seems to be sensible as a way to account for the effect of soft interventions.

The synthetic results are promising. The understanding of what factors affect the casual disentanglement performance is well appreciated.

**Weaknesses:**

The scalability of the method, as acknowledged by the authors, may be a critical concern of the method.

It is not clear if V is supervised and how, or why should p(v) be modeled as a standard Gaussian.

A major concern of the current work, despite its interesting idea and novelty, is the lack of sufficient experimental evaluation to support the understanding of the effectiveness of the presented methodology. Although the results on synthetic data are promising, the results on real data are more ambiguous in providing evidence that the proposed method, specifically the switching variable mechanisms, is effective. Furthermore, other baselines including the explicit LCM models should be included (to demonstrate what is the price being paid without requiring the causal knowledge). The paper in its current form feels rushed and need further work to more thoroughly demonstrate and examine the proposed methodology.

**Questions:**

Regarding my major concern on the experimental outcome, it’d be helpful if the authors could clarify.

For the proposed methodology, what does v represent and is v somehow supervised? It seems that it would be readily available information and including it in some form of supervision would help address the identifiability issue quite a bit.  Also what is h(v) in eq (8)? Is it a regularization term, or is it a learnable function?